# Global Self-Esteem and Stress Intensity in a Group of Polish Nurses—A Mediatory Role of a Sense of Coherence

**DOI:** 10.3390/ijerph19020975

**Published:** 2022-01-15

**Authors:** Ewa Kupcewicz

**Affiliations:** Department of Nursing, Collegium Medicum University of Warmia and Mazury in Olsztyn, 10-719 Olsztyn, Poland; ekupcewicz@wp.pl; Tel.: +48-696-076-764

**Keywords:** stress, global self-esteem, sense of coherence, nurses

## Abstract

(1) Owing to their resistance resources, nurses can reduce the effects of stress, increase their commitment to work and improve their functioning in the face of challenges in the workplace. The aim of this study was to determine the mediatory role of a general sense of coherence and a sense of comprehensibility, manageability and meaningfulness correlated with global self-esteem and the perceived stress intensity in a group of Polish nurses aged 45–55 years. (2) The research using the diagnostic survey method was conducted on a group of 176 nurses (M = 49.1; SD = 3.1) working in seven hospitals located in Olsztyn (Poland). The following were used for data collection: Perceived Stress Scale - PSS-10, Rosenberg’s Self-Esteem Scale and Antonovsky’s Sense of Coherence (SOC-29) Questionnaire. (3) According to 21.02% of the nurses, their stress level at the workplace was low, 44.89% reported it was medium and 34.09% reported it was high. The self-esteem of nearly half of the nurses included in the study (48.30%) was at a medium level, 31.82% felt it was high and 19.89% felt it was low. The mediation analysis showed that a general sense of coherence and a sense of comprehensibility, manageability and meaningfulness have a mediator status in a correlation between global self-esteem and stress intensity. However, their mediatory role is partial. It is desirable for safe work environment promotion programmes to reinforce nurses’ personal resources, which can be helpful in coping with stressors.

## 1. Introduction

According to many studies, the contemporary work environment can be a source of stress and burden for nurses [1,2]. The level of stress perceived by nurses increase while the environmental stressors accumulate. An awareness of occupational stress and the experience of its adverse health consequences encourages prophylactic actions [1,2,3]. There are theoretical reasons and empirical evidence that an important role in the process of coping with adversities is played by one’s personal resources, i.e., a set of relatively stable personal traits, cognitive factors and convictions, which affect the threat assessment and the coping processes [4]. One of such resources is self-esteem, also referred to as a sense of self-worth, which remains in a close relationship with the ability to predict the range of one’s capabilities [5,6,7]. This is extremely important to nurses when they take actions, especially in new, unknown situations that are difficult to manage, because a nurse who starts a new activity should take into consideration her own capabilities. The coping effectiveness also depends on personal resources, referred to as ‘a sense of coherence’. According to Aron Antonovsky, this concept is founded on the assumption that there is a continuum of states between health and disease, which should be perceived as a whole, as a dynamic process of achieving a balance between requirements and resources in confrontation with stress [8].

### 1.1. Stress

The literature mentions various ways of understanding the term, which are not mutually exclusive, but rather complementary. Research into stress increasingly often deals not so much with the experience of stress itself, but with human activities aimed at managing the stressors, referred to as coping with stress [4,9]. The effects of stress are determined to a greater extent by coping with stress than with the objective properties of a stressor [4,9]. It is noteworthy that Steven Hobfoll’s concept has been gaining importance in recent years. It is presented as a resource preservation model, according to which people strive to multiply their resources, understood as items valued by an individual, personal property, circumstances or factors [10]. A literature review confirmed a positive association between personal and professional resources and increased resistance to professional challenges [1,2,11,12]. Many authors included such constructs in a pool of personal and professional resources as: sense of self-efficiency, self-esteem, positive orientation, dispositional optimism, sense of coherence, manageability, social support, work satisfaction and general welfare [1,2,11,12]. According to a review by Yu et al., professional challenges (stress, professional burnout, post-traumatic stress disorder and harassment at work) are negatively correlated with a person’s resistance [12]. Owing to growing resistance resources, nurses can reduce the effects of stress and emotional exhaustion, increase their commitment to work and improve their functioning in the face of challenges at the workplace [12]. In another study conducted by Martos Martínez et al. among Spanish nurses, it was shown that effective coping with stress is affected by the subjective properties of an individual. Some individual variables, such as personality and an affective state can be regarded as stress moderators [13]. Psychological stress has been demonstrated to be associated with somatic health disorders [4]. According to the findings of many studies, mental well-being can be regarded as an indicator of an individual’s adaptation to various critical or crisis events. This happens when the available resources are used in the optimum way to cope with a crisis situation [14,15].

It has been demonstrated in many studies that a shortage of nursing personnel results in work overload, thereby increasing the level of stress and professional burnout among nurses [12]. Measurement of subjectively perceived stress can be used as a good indicator of work overload, associated with the nursing profession [4]. Professional stress prophylaxis has gained special importance as a consequence of the SARS-CoV-2 virus spread. The COVID-19 pandemic has had a great impact on the physical and mental health of healthcare professionals, especially those working in direct contact with patients [16,17].

Many studies have been conducted to seek factors which help one to cope with such crises on an emotional level [18].

### 1.2. Self-Esteem and Health

The definition of self-esteem by Morris Rosenberg is based on the assumption that people have different attitudes towards various objects, and one’s own self is one of such objects [5]. One’s self-image is associated with one’s general well-being, emotions and attitude to tasks [6,7]. The study review shows that self-esteem can be approached in various ways and is sometimes interpreted as a sense of self-worth. According to several studies, people with high self-esteem experience more positive emotions and are more active, tenacious and healthier. On the other hand, people with low self-esteem experience more negative emotions and are less active, and they even show an attitude of avoidance in the face of difficulties, challenges and risk [7]. The first study conducted by Rosenberg showed that people with low self-esteem tend to be seen by nurses at a healthcare centre as more depressive and demonstrating sadness, desolation and disappointment [6,7]. Johnson et al. conducted a study among 306 healthcare professionals in Bangalore to answer the following question: does low self-esteem and a high level of stress lead to professional burnout among healthcare professionals? A path analysis showed that low self-esteem among healthcare professionals had a direct impact on experiencing professional burnout. The impact was also indirect, as employees with low self-esteem were nearly three times more susceptible to high stress levels [19]. The importance of a correlation between self-esteem and professional burnout among nurses was also confirmed in other studies [3,20]. Interesting findings of a study of nurses in Brazil were presented by Santos et al. They found that such factors as smoking, religious beliefs, family income, length of time at work and special events during one’s career could cause accidents and/or changes in employees’ self-esteem, which could threaten their physical and mental health as well as their quality of life and work [21]. Radhakrishnan et al. conducted a study among healthcare professionals working in hospital “front line” settings in India to understand their perceived stigmatising experiences and sense of self-worth during the COVID-19 pandemic [22].

### 1.3. Sense of Coherence

Aaron Antonovsky, an eminent medical sociologist, claimed that a mental factor existed which had a positive impact on mechanisms of coping with stress and had a key role in processes of achieving, protecting and restoring good health. He called this factor ‘a sense of coherence’ (SOC) [8,23,24]. According to Antonovsky, SOC is a global, complex orientation of a person, showing the degree of conviction that the stimuli reaching the person’s body from the inner and outer environment are structured, predictable and ones that can be explained; resources are available which allow for meeting the requirements set by these stimuli and it is worth the commitment and taking actions to meet the requirements [8]. SOC is regarded as a meta-resource, on which the utilisation of other resources largely depends. It comprises a sense of comprehensibility, manageability and meaningfulness that a person has [8,24]. The comprehensibility component concerns a cognitive aspect of the situation—it is possible to work out and explain reality. The sense of manageability is referred to as an instrumental component—an individual has a sense of having resources at their disposal to cope with requirements. Meaningfulness is the degree of feeling the sense of life. Being an emotional–motivational component, it gives one a sense that it is worth engaging oneself in an action [8,23,24]. Antonovsky’s theory of salutogenesis shows that human health can be affected by individual and work-related factors [8,23,24]. Masanotti et al. conducted a wide-ranging literature review and found SOC to be a predictor of depression, professional burnout and dissatisfaction with work among nurses [25,26,27]. The findings of a study conducted by Betke et al. confirmed that the sense of coherence promotes health in a stressful work environment – a high sense of coherence in nurses translates into better mental health, proper functioning in the work environment and following adaptative strategies of coping with stress [28]. Another study demonstrated a diversity in the three sense of coherence dimension levels among nurses in Sweden. The lowest level was observed in the manageability component, and the highest was in meaningfulness [29].

The aim of this study was to determine the mediatory role of a general sense of coherence and a sense of comprehensibility, manageability and meaningfulness in a correlation between global self-esteem and the perceived stress intensity in a group of Polish nurses aged 45–55 years.

## 2. Materials and Methods 

### 2.1. Settings and Design

A study using the diagnostic survey method was conducted among a group of 176 nurses working between June 2013 and January 2015 in seven hospitals located in Olsztyn, in the Warmińsko-Mazurskie Voivodship (Poland).

The group included women who had not used hormonal replacement therapy during the past six months and gave informed consent to participate in the study. The exclusion criteria included menopause caused by hazardous chemical, radioactive factors, surgery or other external factors and a failure to provide consent to participate in the study. After the management of all hospitals gave their consent to conduct the study, the researcher (E.K.) personally delivered the questionnaire packages to the sites. The subjects were informed about the study objective and had an opportunity to ask questions and receive answers. After the nurses gave their consent to participate in the study, 201 packages of questionnaires were provided. It took about 20 minutes to complete the questionnaire. The participation was voluntary, anonymous, and the respondents were allowed to resign at any time without giving a reason. After the data were collected and after incomplete questionnaires were eliminated, 176 questionnaires (87.56%) were taken for further statistical analysis. 

The study was performed in accordance with the Helsinki Declaration, and a favourable opinion was given by the Senate Research Ethics Committee of Olsztyn University College J. Rusiecki (No.11/2016), Poland. This met the criteria of a cross-sectional study, and it is part of a greater research project carried out among a group of nurses in Poland [30,31].

### 2.2. Research Instruments

The diagnostic survey method was applied, and the following validated diagnostic tools in the Polish language versions were used to collect data:

Perceives Stress Scale - PSS-10 by S. Cohen, T. Kamarck, R. Mermelstein in a Polish adaptation by Z. Juczyński, N. Ogińska-Bulik [4];

Self-Esteem Scale (SES) developed by M. Rosenberg, in a Polish adaptation by I. Dzwonkowska, K. Lachowicz-Tabaczek, M. Łaguna [6];

Sense of Coherence (SOC-29) Questionnaire by A. Antonovsky, in a Polish adaptation by J. Koniarek, B. Dudek, Z. Makowska [32];

The respondent group was characterised by the author’s original questionnaire, which contained questions about the respondents’ age, education, place of residence, marital status, financial situation, frequency of gynaecological check-ups and forms of physical exercise.

#### 2.2.1. Perceived Stress Scale (PSS-10)

This scale is used to assess the intensity of stress associated with one’s life situation over the past month. The stress intensity is determined not by a number of events but rather by their assessment. The scale contains ten questions about various subjective feelings related to personal problems and events, behaviours and methods of coping. For each question, it was a respondent’s task to answer how often they thought and felt like that. Answers are provided a 5-point scale by assigning 0 to 4 points to each response, according to the following rules: 0 = never, 1 = hardly ever, 2 = sometimes, 3 = quite often, 4 = very often. The scores for answers to positive questions, i.e., 4, 5, 7, 8, were changed before the overall index of perceived stress was calculated, according to the rule: 0 = 4; 1 = 3; 3 = 1; 4 = 0. The overall score was the sum of all points, with a theoretical distribution from 0 to 40. A higher score reflected a higher perceived stress level. The internal reliability of the PSS-10 scale, evaluated on the basis of Cronbach alpha, is 0.86 [4].

#### 2.2.2. Rosenberg’s Self-Esteem Scale (SES)

Rosenberg’s Self-Esteem Scale is used for measuring the global explicit self-esteem, regarded as a relatively permanent attitude towards oneself. It is made up of ten diagnostic statements. A respondent was asked to indicate the extent to which he/she agreed with each of the statements by circling one answer. The answers are given on a 4-degree scale, from 1 to 4 (1—I definitely agree, 4—I definitely disagree). The respondent received from 1 to 4 points for each answer. With the adopted method of assessment, positive statements were reversed: 1, 2, 4, 6, 7 so that a higher score was given for answers which expressed a higher self-esteem level. A respondent could receive a score of 10 to 40 points. A higher score reflected higher self-esteem. The SES scale has good psychometric properties. The questionnaire reliability, estimated with Cronbach’s alpha for various standardisation groups, was satisfactory between 0.81 and 0.83 [6].

#### 2.2.3. Sense of Coherence (SOC-29) Questionnaire 

Developed by Aaron Antonovsky, the Sense of Coherence questionnaire comprises 29 test items, expressed as questions about various aspects of human life. It can be used to estimate a general level of the sense of coherence and the levels of its three components, i.e., comprehensibility (11 statements), manageability (ten statements) and meaningfulness (eight statements). The respondents assess the accuracy of each statement with respect to themselves and their own lives, using a 7-point Likert’s scale, where “1” means that the attitude always occurs, whereas “7” indicates that it never does. The points are subsequently summed up, and a higher score reflects a higher level of each component and a higher overall sense of coherence. The overall score in the SOC-29 questionnaire can range from 7 to 203 points. The SOC-29 questionnaire has good psychometric properties. In accordance with the theoretical assumptions, significant correlations were observed with various health parameters, and the high reliability of the questionnaire was demonstrated. The internal consistency indices calculated by the split-half method with the Spearman–Brown correction were: 0.92 for the overall sense of coherence, 0.78 for comprehensibility, 0.72 for manageability and 0.68 for meaningfulness [32].

### 2.3. Statistical Analysis

The data were analysed with the Polish version of STATISTICA 13 (TIBCO, Palo Alto, CA, USA). Demographic data are shown as the number of cases and %. The following were used to describe the variables: arithmetic mean (M), standard deviation (SD), minimum (min.) and maximum (max.), 95% confidence interval (CI). The indices of the overall global self-esteem and of the perceived stress intensity were transferred into standardised units (score range 1–10), interpreted according to the properties characterising the sten scale. A score between 1 and 4 sten was regarded as low, whereas those from 7 to 10 sten were regarded as high, which corresponds to the area of approx. 33% of observations. A score of 5 and 6 sten was regarded as average [4,6]. The mediatory effects were measured by the methodology developed by R.M. Baron and D.A. Kenny [33]. The Sobel test was used to verify the statistical significance of the mediation analysis model [34].

The correlation power interpretation was based on the Guilford classification, as follows: |r| = 0—no correlation; 0.0 < |r| ≤ 0.1—slight correlation; 0.1 < |r| ≤ 0.3—poor correlation; 0.3 < |r| ≤ 0.5—average correlation; 0.5 < |r| ≤ 0.7—high correlation; 0.7 < |r| ≤ 0.9—very high correlation; 0.9 < |r| < 1.0—nearly full correlation; |r| = 1—full correlation [35]. Values of *p* < 0.05 were regarded as statistically significant.

## 3. Results

### 3.1. Participants

A total of 176 nurses aged 45–55 years (M = 49.1; SD = 3.1) took part in the study. Nearly half of them (*n* = 98; 55.68%) had secondary education (they had completed a secondary medical school or a post-secondary vocational school). A great majority of the respondents lived in a town (*n* = 153; 86.93%), were married or lived in a partnership (*n* = 143; 81.25%). According to 46.02% of the respondents, their financial situation was satisfactory, whereas 40.34% (*n* = 71) claimed that it was good or very good (Table 1).

### 3.2. The Stress Intensity and Global Self-Esteem

The assessment of the stress intensity on the PSS-10 scale aimed at identifying the subjective feelings of the study subjects associated with personal issues and events and the methods of coping with them. It was shown that the mean perceived stress level was 17.4 (SD = 5.3) on a scale of 0 to 40 points over the past month. The detailed results of the descriptive statistics for stress intensity and global self-esteem in the study group are shown in Table 2.

Transformation of the overall perceived stress intensity index into standardised units resulted in demonstrating that every fifth woman (21.02%) in the study group experienced a low stress level in association with her life situation during the past month, whereas 44.89% of them perceived the stress intensity as average and 34.09% perceived the stress intensity as high (Figure 1). 

Subsequently, an analysis was performed of the global self-esteem level—a relatively constant disposition understood as a conscious (positive or negative) attitude towards oneself. The mean result for the whole sample was 30.9 (SD = 4.2), with a range between 10 and 40 points (Table 1). When transformed into standardised units on the sten scale, the results show that the self-esteem of nearly half of the nurses included in the study (48.30%) was at an average level, 31.82% at a high level and 19.89% at a low level (Figure 1).

### 3.3. The Sense of Coherence

According to Antonovsky’s theory of salutogenesis, the sense of coherence among nurses was analysed as a generalised, emotional and cognitive outlook on the world, containing three inextricably linked components: comprehensibility, manageability and meaningfulness. The statistical analysis showed that the mean sense of coherence index in the group of nurses under study aged 45–55 years was 130.5 points (SD = 19.8) on a scale from 7 to 203. The mean result for comprehensibility, manifesting itself as an ability to understand the surrounding world, was 47.0 points (SD = 10.6) on a scale from 7 to 77. The mean result for the manageability component, showing that individuals are confident of being able to cope with a situation they find themselves in owing to the resources at their disposal, was 47.5 (SD = 7.8) on a scale from 7 to 70 points. The third component, called meaningfulness, which decides that it is worth getting engaged in various important and valuable aspects of life, reached a mean score of 36.0 points (SD = 5.5) on a scale from 7 to 56. Table 3 shows the results for descriptive statistics for the global sense of coherence and its three components in the group of nurses under study. 

### 3.4. Mediation Analysis

A mediation analysis was performed in order to check whether a general sense of coherence and a sense of comprehensibility, manageability and meaningfulness are significant mediators in a correlation between global self-esteem and the perceived stress intensity in the nurses under study. The following assumptions were adopted in the developed models:

path a = a correlation between an independent variable and a mediator,

path b = a correlation between a mediator and a dependent variable (with the independent variable control),

path c = a direct correlation between an independent variable and a dependent variable,

path c’ = a correlation between an independent variable and a dependent variable (with a mediator control).

#### 3.4.1. The Correlation between Global Self-Esteem and the Stress Intensity with a Mediator—A General Sense of Coherence

The first step of the mediation analysis involved checking whether the independent variable—global self-esteem—is correlated to the mediator—a general sense of coherence (without the dependent variable – stress intensity). This correlation proved to be high and positive, and statistically significant: F(1, 174) = 70.28; *p* < 0.0001, *β* = 0.536; *p* < 0.0001. Global self-esteem was adopted as a predictor of a sense of coherence. 

The second step involved checking the correlation between an independent variable and a dependent variable, without a mediator. The research model developed for the purpose proved to be well-fitted to the data: F(1, 174) = 67.96; *p* < 0.0001, *β* = −0.529; *p* < 0.0001 and indicated a highly negative correlation between the global self-esteem and the stress intensity. This means that the higher the coefficient of regression between the variables, the higher the probability of causality. 

A mediator was introduced to the model in the third step, and the significance of the mediation model was evaluated. The power of correlation between an independent variable and a dependent variable after a mediator was introduced decreased, but it is still statistically significant (*β* = −0.297; *p* < 0.0001), and the mediator proved to be highly negatively correlated to the dependent variable: F(1, 174) = 94.57; *p* < 0.0001, *β* = −0.593; *p* < 0.0001. This means that a general sense of coherence is a partial mediator of a correlation between global self-esteem and stress intensity. The analysis demonstrated significant mediation, confirmed by the Sobel test result (z = 3.87; *p* < 0.0001) (Figure 2).

#### 3.4.2. Correlation between Global Self-Esteem and the Stress Intensity with a Mediator—A Sense of Comprehensibility

In the next step of statistical analyses, actions were taken to verify the assumption of the mediatory role of a sense of comprehensibility in the correlation between global self-esteem and stress intensity. The first step in developing a mediation model involved confirming a significant, positive correlation of the average power between an independent variable and a mediator: F(1, 174) = 33.08; *p* < 0.0001, *β* = 0.399; *p* < 0.0001. 

The second step involved testing a direct correlation between an independent variable and a dependent variable. The correlation between global self-esteem and stress intensity proved to be statistically significant and the model developed for the purpose proved to be well-fitted to the data: F(1, 174) = 67.96; *p* < 0.0001, *β* = −0.529; *p* < 0.0001. 

A variable described as a mediator—a sense of comprehensibility—was introduced to the model in the third step. The role of an independent variable in a model which included both a mediator and an independent variable decreased, but it remains statistically significant (*β* = −0.407; *p* < 0.0001), whereas the mediator proved to be negatively correlated on an average level to the dependent variable: F(1, 174) = 49.26; *p* < 0.0001, *β* = −0.470; *p* < 0.0001. The result indicating a mediation was confirmed with the Sobel test (z = 4.49; *p* < 0.0001) (Figure 3).

#### 3.4.3. A Correlation between the Global Self-Esteem and the Stress Intensity with a Mediator—A Sense of Manageability

The mediatory role of a sense of manageability between global self-esteem and stress intensity was examined at a further stage. A positive correlation between global self-esteem and a sense of manageability was demonstrated in the first step: F(1, 174) = 42.55; *p* < 0.0001, *β* = 0.443; *p* < 0.0001, and in the second step this was followed by a confirmation of a direct and negative correlation between global self-esteem and the stress intensity: F(1, 174) = 67.96; *p* < 0.0001, *β* = −0.529; *p* < 0.0001. 

A variable described as a mediator—a sense of manageability—was introduced to the model in the next step. After a mediator—a sense of manageability—and an independent variable was taken into account in the analyses, the power of correlation between the global self-esteem and intensity of perceived stress decreased, but it remains statistically significant (*β* = −0.356; *p* < 0.0001), whereas the mediator proved to be highly negatively correlated with the dependent variable: F(1, 174) = 74.95; *p* < 0.0001, *β* = −0.549; *p* < 0.0001. The result indicating a mediation was confirmed with the Sobel test (z = 4.84; *p* < 0.0001) (Figure 4).

#### 3.4.4. A Correlation between Global Self-Esteem and the Stress Intensity with a Mediator—A Sense of Meaningfulness

The aim of the last mediation analysis was to determine whether a sense of meaningfulness plays a significant mediatory role between global self-esteem and the stress intensity in the nurses under study. 

The first step involved testing a correlation between global self-esteem and a sense of meaningfulness, which is positive and statistically significant: F(1, 174) = 55.05; *p* < 0.0001, *β* = −0.490; *p* < 0.0001. A direct negative correlation between global self-esteem and the stress intensity was confirmed in the second step: F(1, 174) = 67.96; *p* < 0.0001, *β* = −0.529; *p* < 0.0001. A mediator—a sense of meaningfulness—was introduced to the model in the third step, and the significance of the mediation model was evaluated. The power of correlation between an independent variable and a dependent variable after a mediator—a sense of meaningfulness—was introduced decreased slightly, but it is still statistically significant (*β* = −0.443; *p* < 0.0001), and the mediator proved to be significantly correlated to the dependent variable: F(1, 174) = 31.82; *p* < 0.0001, *β* = −0.393; *p* < 0.0001. One can conclude that a sense of meaningfulness is a partial mediator of a correlation between global self-esteem and stress intensity. The mediation in the model under discussion was confirmed by the Sobel test result (z = 4.26; *p* < 0.0002) (Figure 5).

## 4. Discussion

The mean stress intensity score on the PSS-10 scale in this study was 17.40 (SD = 5.3), and it was slightly higher than in the normalisation study in a group of healthy subjects [4]. A comparison of the score between this study and the results for nursing students in an international study showed that the mean stress intensity levels for the nursing students were higher in the group in Poland, in Spain and in Slovakia than in the group of nurses aged 45–55 years in this study [36]. It is also noteworthy that a Polish normalisation study revealed statistically significant differences in the stress intensity in clinical groups. Relatively low stress levels were noted in patients on dialysis. The mean perceived stress level on the PSS-10 scale in this group was 16.87 (SD = 5.55), and it was lower than in the group of nurses in this study. A high perceived stress intensity was observed in postmenopausal women and in people after a cardiac infarction, which is a clear indication that they should be provided with special psychological care [4]. The data from the current study showed that 34.09% of the nurses experienced high levels of stress during the past month associated with their own life situation, which may indicate that they are excessively loaded with their work as nurses. Other studies show that high stress levels in nursing students cause a decrease in the quality of their lives [36]. Zheng et al. argue that nurses are exposed to more serious effects of stress, including health-related effects, during the COVID-19 pandemic [37]. Mazza et al., who conducted a study in Italy, confirmed that the effects of the SARS-CoV-2 virus, associated, for example, with the quarantine, bring about an increased level of stress, anxiety and depression in the general population [38].

Global self-esteem, manifesting itself in how one thinks about oneself, is another variable under analysis. The current study revealed the mean global self-esteem score for the whole group to be 30.9 points and was comparable with the mean calculated from the data collected in 53 countries (M = 30.85) [39]. The study results show that the self-esteem of nearly half of the nurses included in the study, aged 45–55 years (48.30%), was at an average level, 31.82% reported a high level, and 19.89% reported a low level. A different study in Poland produced similar findings [3]. An analysis of the correlations of global self-esteem with different variables should address a relatively new trend in research of a sympathetic attitude towards oneself, proposed by Kristin Neff, who understands it as “being open and sensitive to one’s suffering, while experiencing care and kindliness towards oneself, adopting an understanding and non-judging attitude towards one’s flaws and failures, realising that one’s own experience is part of the common experience of mankind” [40]. The findings of a study conducted by Dzwonkowska showed that people who are more sympathetic towards themselves, with higher self-esteem, are less susceptible to depression and they feel fewer negative emotions and are less lonely, and at the same time experience more positive affective states than those less sympathetic and with lower self-esteem [41]. The feeling of self-worth is an important source of coping with stress and health protection, and it is associated with work satisfaction. Studies of the correlations between the stress intensity and the coping strategies followed by nurses confirm a correlation between these two variables. Duran et al. conducted a study among a group of 121 nurses in Turkey who worked at intensive care units and showed that those with high self-esteem have fewer psychological issues than those with low self-esteem [42]. In their study, Dimunová et al. found the choice of a coping strategy by nurses working in anaesthesiology and intensive care departments in Slovakia to be correlated with self-esteem. Nurses with high self-esteem levels preferred adaptive coping strategies, such as active coping, positive redefinition, planning and acceptance. On the other hand, nurses with average and low global self-esteem scores preferred non-adaptive coping strategies: denial, lack of commitment and blaming themselves [43].

Performing the nursing profession entails frequent difficult (often stressful) situations as a consequence of being responsible for the highest values—human health and life. According to many researchers, a sense of coherence can considerably contribute to maintaining good health, modify one’s functioning in a stressful work environment, protect nurses from professional burnout and affect the choice of a strategy for coping with stress [23,25,28]. The analyses performed in the current study show that the mean score for the overall sense of coherence for nurses aged 45–55 years was 130.5 (SD = 19.8) points, for comprehensibility—47.0 (SD = 10.6), for manageability—47.5 (SD = 7.8) and for meaningfulness—36.0 (SD = 5.5). These findings do not differ significantly from other studies conducted among Polish nurses [44]. Kocięcka et al. found nurses with high SOC scores to exhibit fewer somatic disorders, anxiety, sleep disorders and to be less prone to depression [45]. Similarly, a study conducted by Malagon-Aguilera among a group of 109 nurses working in a long-term medical care facility found that nurses with a higher sense of coherence were healthier and more committed to their work [46]. Ando et al. conducted a study of a group of 130 psychiatric nurses in a hospital in Japan and found that the sense of coherence affected their work satisfaction and was negatively correlated with unethical behaviour [47]. A study conducted by Jażdżewska et al. showed an increase in the hope for success was accompanied by an increase in the sense of coherence in the nurses [48]. A different study of a group of women with breast cancer being treated with chemotherapy showed women with a high sense of coherence reported lower fatigue, lower incidence of adverse effects of chemotherapy, and they enjoyed a better quality of life [49]. Zamanian et al. conducted a study of a group of 221 women with breast cancer and observed a positive correlation between the sense of coherence and quality of life. They concluded that practical activities aimed at reinforcing the sense of coherence in women should be included in rehabilitation programmes [50]. According to Colomer-Pérez et al., reinforcing the sense of coherence in nurse assistants through a salutogenic education strategy increases their health resources and contributes to higher resistance to work-related stress [51].

The findings of a study by Allan et al. showed that the level of perceived stress at work had a significant, negative correlation with a sense of life. Understanding the importance of one’s personal development at work improves the protection against the negative impact of occupational stress [52]. Occupational stress is an inevitable, sometimes essential, element of the work environment, but it does not have to result in an organisational, psychological, behavioural or medical dysfunction [53].

In conclusion, it should be noted that global self-esteem and a sense of coherence as subjective individual properties make nurses—even if they perceive some events as stressors—cope with them better and will not experience negative effects of the stress transaction (or will experience them as weaker).

### Limitations and Implications for Professional Practice

It has been demonstrated in this study that global self-esteem is a predictor for a sense of coherence and a sense of comprehensibility, manageability and meaningfulness. The analysis has shown that the higher the self-esteem level, the higher the conviction of a sense of coherence. A significant negative correlation has also been found between global self-esteem and stress intensity. Moreover, it has been confirmed that a general sense of coherence and its components have a mediator status in a correlation between global self-esteem and the stress intensity in the group of nurses aged 45–55 years. It has been revealed that the components of a sense of coherence constitute the essential cognitive and emotional–motivational resources which determine one’s capability for coping with stress.

At the same time, they do not complete the research of the correlations discussed here. The study author indicated certain limitations, as the study group comprised nurses of only one age group. It would also be worthwhile to perform analyses taking into consideration gender as a variable.

The findings may be useful in adapting psychological interventions oriented towards styles and strategies of coping with stress by nurses in their work environments and in private lives. There are also reasons to develop and perfect the nurses’ abilities to build a positive self-attitude, and a sense of comprehensibility, manageability and meaningfulness. 

## 5. Conclusions

More than 1/3 of the nurses in the study group experienced high-intensity stress during the past month.

The global self-esteem score for approx. 80% of the nurses was average or high.

The results for the overall sense of coherence and its components: comprehensibility, manageability and meaningfulness, do not differ much from those in other studies conducted among Polish nurses.

It has been shown in the adopted mediation model that a general sense of coherence and a sense of comprehensibility, manageability, and meaningfulness have a mediator status in a correlation between global self-esteem and stress intensity. However, their mediatory role is partial in the correlation between the variables.

It is desirable for safe work environment promotion programmes to reinforce nurses’ personal resources, which can be helpful in coping with stressors.

## Figures and Tables

**Figure 1 ijerph-19-00975-f001:**
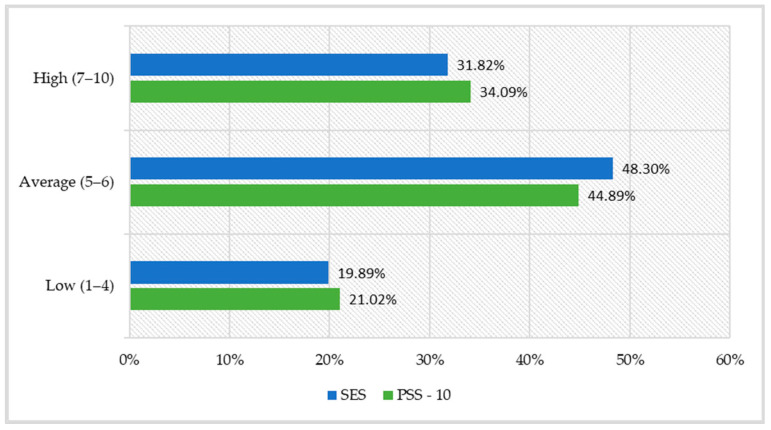
Distribution of global self-esteem scores and the level of perceived stress among the nurses in the study.

**Figure 2 ijerph-19-00975-f002:**
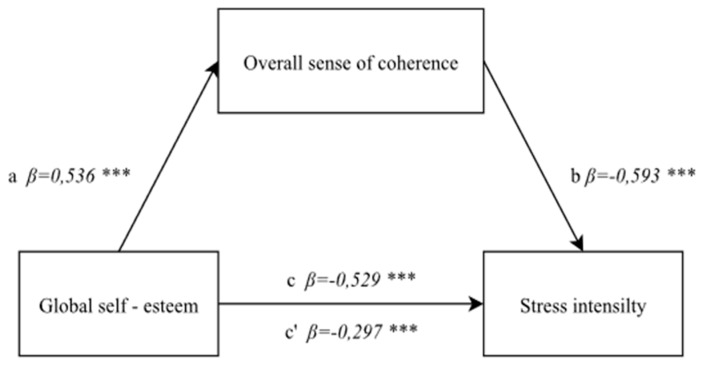
Model of a mediatory role of a general sense of coherence between global self-esteem and stress intensity. Statistically significant: *** *p* < 0.001.

**Figure 3 ijerph-19-00975-f003:**
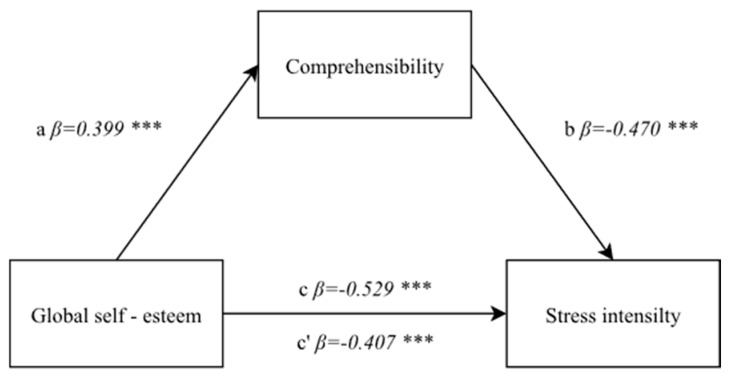
A model of the mediatory role of a sense of comprehensibility between self-esteem and the stress intensity. Statistically significant: *** *p* < 0.001.

**Figure 4 ijerph-19-00975-f004:**
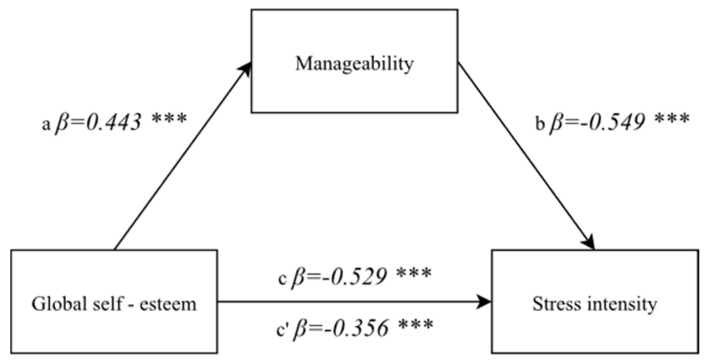
A model of the mediatory role of a sense of manageability between global self-esteem and the stress intensity. Statistically significant: *** *p* < 0.001.

**Figure 5 ijerph-19-00975-f005:**
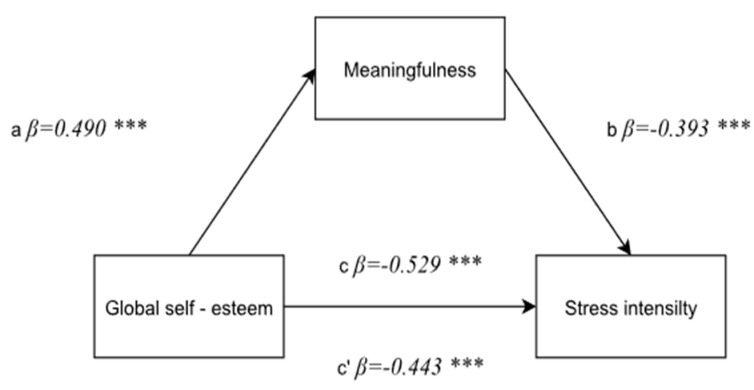
A model of the mediatory role of a sense of manageability between global self-esteem and the stress intensity. Statistically significant: *** *p* < 0.001.

**Table 1 ijerph-19-00975-t001:** Characterisation of the group under study (*n* = 176).

Variables		Frequency (*n*)	Percentage (%)
Education	secondary	98	55.68
	university	78	44.32
Place of residence	village	23	13.07
	town	153	86.93
Marital status	single	11	6.25
	married/in a civil relationship	143	81.25
	widow	4	2.27
	divorced/separated	18	10.23
Financial status	good/very good	71	40.34
	satisfactory	81	46.02
	poor/very poor	24	13.64

**Table 2 ijerph-19-00975-t002:** Results of descriptive statistics for stress intensity, global self-esteem and sense of coherence in the study group.

Variables	*n* = 176			
	M	95% Cl	Min.–Max.	SD
PSS-10	17.4	16.6–18.2	0–33.0	5.3
SES	30.9	30.3–31.6	19.0–40.0	4.2

*n*—number of subjects; PSS-10—stress intensity, SES—global self-esteem, SOC-29—overall sense of coherence, M—arithmetic mean; 95% confidence interval (CI); Min.—minimum; Max.—maximum; SD—standard deviation.

**Table 3 ijerph-19-00975-t003:** Results of descriptive statistics for the sense of coherence in the study group.

Variables		*n* = 176			
		M	95% Cl	Min.–Max.	SD
SOC-29 (7–203 points)		130.5	127.5–133.4	71.0–188.0	19.8
SOC-29 components	Comprehensibility (7–77 points)	47.0	45.4–48.6	27.0–67.0	10.6
	Manageability (7–70 points)	47.5	46.4–48.7	9.0–67.0	7.8
	Meaningfulness (7–56 points)	36.0	35.1–35.8	18.0–50.0	5.5

*Explanation:**n*—number of subjects; SOC-29—overall sense of coherence, M—arithmetic mean; 95% confidence interval (CI); Min.—minimum; Max.—maximum; SD—standard deviation.

## Data Availability

The data presented in this study are available on request from the corresponding author.

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
