# Peer review of "Global Self-Esteem and Stress Intensity in a Group of Polish Nurses—A Mediatory Role of a Sense of Coherence"

_ijerph, 2022, doi:10.3390/ijerph19020975_

Round 1
Reviewer 1 Report
I am grateful for the opportunity to review the interesting topic “Stress intensity and global self-esteem as variables determining the sense of coherence among Polish nurses aged 45-55 years ”.
The following recommendations are designed to improve the manuscript:
Lines 52-53: As we know, psychological distress can lead not only to somatic health disorders (such as somatization disorder), but also to mental health problems (such as anxiety, depression). It is recommended that the authors of the paper clarify the relationships between psychological stress and mental well-being.
Lines 136-137: Maybe the authors of the manuscript could indicate the issue number provided by the Senate Research Ethics Committee of Olsztyn University College.
Lines 140-150: The “Participants” part of the manuscript contains the results (characterizes the seciodemographic condition of respondents). I would consider the authors must move the aforementioned description to the Results of the manuscript.
Line 208: I would suggest that “level of confidence of the mean (Cl ± 95%)” be replaced by “a 95% confidence interval (CI) (95% CI)”.
Lines 233, 269: 1-2 Tables of the manuscript refer to mean and median. Means must be reported when the distribution of the data meets the normal conditions (data are normally distributed) and medians when the distribution of the data does not meet the normality. I would therefore suggest that authors of paper may to choose: either mean ± SD (standard error) or median ± SE (standard deviation).
Figure 1 and Figure 2 are one-dimensional. The information provided in the Figures (1-2) overlapes the information of the manuscript text. I would suggest considering the necessity of providing figures 1-2.
Lines 226, 254: It is advisable to renounce part of the title “– distribution of results” and potentially write: “The Stress Intensity and Global Self-esteem” and “The Sense of Coherence.”
Line 272: I would suggest that the title “An analysis of the correlation between the stress intensity, the global self-esteem and the sense of coherence in the study group” could be shortened accordingly: “The Correlation Between the Stress Intensity, the Global Self-esteem and the Sense of Coherence”.
Lines 274-277, 298-302: The information on these lines is redundant and could be moved to the Methods.
Lines 422-428: I would suggest that the text in these lines could complement conclusions. According to this context, the authors of the manuscript may to globalize the findings and justify the necessary practical recommendations.
Lines 430-440: I would suggest merging conclusions into a single paragraph.
Lines 9-25: I would suggest to delete words such as Background and Objectives, Materials and Methods, Results, Conclusions. In addition, the abstract consists of 262 words and is too long. According to the provisions of the IJERPH, the abstract should be a total of about 200 words maximum.
In general, the authors prepared a well-written paper. After minor changes, I recommend to accept the manuscript.
Best Regards
Author Response
Dear Reviewers, Thank you very much for a thorough editorial assessment of my manuscript, positive opinions, as well as the reviewers’ remarks. I used them as an important guide to improving the quality of my paper. The corrections were implemented strictly according to their comments. All changes made in the text are marked in yellow and blue. I have enclosed the re-edited manuscript and cover letter as responses to Reviewers, detailing how I followed their suggestions. Thank you very much for your kind consideration of my paper. Yours sincerely, Ewa Kupcewicz, PhD
|
Lp. |
Reviewer 1 |
Responses
|
|
1. |
Lines 52-53: As we know, psychological distress can lead not only to somatic health disorders (such as somatization disorder), but also to mental health problems (such as anxiety, depression). It is recommended that the authors of the paper clarify the relationships between psychological stress and mental well-being. |
The title of the paper has been altered. The aim of the paper has been modified, and every effort has been made to make the "Introduction" section more valuable. The author of the paper indicated that ”psychological well-being can be recognised as an indicator of an individual's adaptation to various critical or crisis events" and explained why this is the case. |
|
2. |
Lines 136-137: Maybe the authors of the manuscript could indicate the issue number provided by the Senate Research Ethics Committee of Olsztyn University College. |
Reference has been given to the approval of the Senate Research Ethics Committee number (No. 11/2016). |
|
3. |
Lines 140-150: The “Participants” part of the manuscript contains the results (characterizes the seciodemographic condition of respondents). I would consider the authors must move the aforementioned description to the Results of the manuscript. |
The subsection ”Participants" was moved to the ”Results" section, and the sociodemographic characteristics of the respondents are presented in a table. |
|
4. |
Line 208: I would suggest that “level of confidence of the mean (Cl ± 95%)” be replaced by “a 95% confidence interval (CI) (95% CI)”. |
The changes were made according to suggestions |
|
5. |
Lines 233, 269: 1-2 Tables of the manuscript refer to mean and median. Means must be reported when the distribution of the data meets the normal conditions (data are normally distributed) and medians when the distribution of the data does not meet the normality. I would therefore suggest that authors of paper may to choose: either mean ± SD (standard error) or median ± SE (standard deviation). |
The changes were made according to suggestions |
|
6. |
Figure 1 and Figure 2 are one-dimensional. The information provided in the Figures (1-2) overlapes the information of the manuscript text. I would suggest considering the necessity of providing figures 1-2. |
One new figure was presented, which shows the results of the sten scale test for global self-esteem and stress intensity. |
|
7. |
Lines 226, 254: It is advisable to renounce part of the title “– distribution of results” and potentially write: “The Stress Intensity and Global Self-esteem” and “The Sense of Coherence.” Line 272: I would suggest that the title “An analysis of the correlation between the stress intensity, the global self-esteem and the sense of coherence in the study group” could be shortened accordingly: “The Correlation Between the Stress Intensity, the Global Self-esteem and the Sense of Coherence”. |
The subsection titles were modified as recommended. However, the subsection, ”An analysis of the correlation between the stress intensity, the global self-esteem and the sense of coherence in the study group", was completely removed. A new subsection , "Analysis of mediation", was introduced. |
|
8. |
Lines 274-277, 298-302: The information on these lines is redundant and could be moved to the Methods. |
The indicated lines were removed. |
|
9. |
Lines 422-428: I would suggest that the text in these lines could complement conclusions. According to this context, the authors of the manuscript may to globalize the findings and justify the necessary practical recommendations. |
The indicated lines were removed. |
|
10. |
Lines 422-428: I would suggest that the text in these lines could complement conclusions. According to this context, the authors of the manuscript may to globalize the findings and justify the necessary practical recommendations. Lines 430-440: I would suggest merging conclusions into a single paragraph. |
The subsections “Limitations and implications for professional practice” and “Conclusions” were modified.
|
|
11. |
Lines 9-25: I would suggest to delete words such as Background and Objectives, Materials and Methods, Results, Conclusions. In addition, the abstract consists of 262 words and is too long. According to the provisions of the IJERPH, the abstract should be a total of about 200 words maximum. |
The abstract of the paper was modified and adapted to IJERPH requirements. |
|
12. |
As suggested by the Reviewer, final linguistic corrections were carried out by the language translation office - OSCAR - Foreign Language School and Translation Office Joanna Jensen located at ul. Reja 2/4 lok 1, 10-565 Olsztyn, Poland. |
|
Reviewer 2 Report
The strong point of this study is the researcher's choice to investigate the relationship between sense of coherence and self-esteem and it is true that there is little literature on this subject. Unfortunately the good news stops here, the manuscript presents serious problems.
-The first very important problem is that the author uses the sense of coherence as a state of personality while it is a trait part of personality. “The SOC is often considered to be a stable entity that develops in young adulthood and stabilizes around the age of 30 “ [Monica Eriksson and Maurice B. Mittelmark. The Handbook of Salutogenesis [Internet]. Chapter 12; The Sense of Coherence and Its Measurement ; https://www.ncbi.nlm.nih.gov/books/NBK435831/].
It is therefore not possible for stress, which is changing, to determine a pre-existing fixed personality trait. I would ask the author to rewrite the manuscript trying to determine the Stress intensity based on the sense of coherence and self-esteem.
-The second important problem of the study is the sample, the participants in the research. The participants are exclusively women of a certain age group. Is there a good reason for such a choice? The author should state the reason for choosing such a specific group.
-The third issue is related to the statistical analysis of the manuscript to which mediation analysis must be added in order to support a model of possible causation. Please also consider whether the sample meets the requirements for regression analysis.
In closing, I would like to ask for some minor corrections:
- Delete some information that seems irrelevant to the study such as:
Line 148-150 (About 3/4 of the respondents had a gynaecological check-up once a year, whereas the others reported that they had not visited a gynaecologist for a check-up for over two years). Line 162 (frequency of gynaecological check-ups and forms of physical exercise).
- Add the time period in which the study was conducted, it matters, especially for stress whether it was conducted before or during the pandemic.
Author Response
Dear Reviewers, Thank you very much for a thorough editorial assessment of my manuscript, positive opinions, as well as the reviewers’ remarks. I used them as an important guide to improving the quality of my paper. The corrections were implemented strictly according to their comments. All changes made in the text are marked in yellow and blue. I have enclosed the re-edited manuscript and cover letter as responses to Reviewers, detailing how I followed their suggestions. Thank you very much for your kind consideration of my paper. Yours sincerely, Ewa Kupcewicz, PhD
|
Lp. |
Reviewer 2 |
Responses
|
|
1. |
The strong point of this study is the researcher's choice to investigate the relationship between sense of coherence and self-esteem and it is true that there is little literature on this subject. Unfortunately the good news stops here, the manuscript presents serious problems. -The first very important problem is that the author uses the sense of coherence as a state of personality while it is a trait part of personality. “The SOC is often considered to be a stable entity that develops in young adulthood and stabilizes around the age of 30 “ [Monica Eriksson and Maurice B. Mittelmark. The Handbook of Salutogenesis [Internet]. Chapter 12; The Sense of Coherence and Its Measurement ; https://www.ncbi.nlm.nih.gov/books/NBK435831/]. It is therefore not possible for stress, which is changing, to determine a pre-existing fixed personality trait. I would ask the author to rewrite the manuscript trying to determine the Stress intensity based on the sense of coherence and self-esteem. |
The title of the paper has been altered. The aim of the study was modified and efforts were made to make the "Introduction" section more valuable. An in-depth analysis of the literature was carried out and the "Introduction" section was clarified with regard to the interpretation of the examined variables (global self-esteem, sense of coherence and stress intensity). Self-esteem in Rosenberg's approach is treated as a trait, i.e. a disposition that is relatively stable over time. The sense of coherence is subject to development during the individual's life. The position on the continuum of the sense of coherence becomes relatively fixed at the threshold of adulthood, when inconsistencies in various areas of life - prevalent in adolescence - are sorted out or accepted (A. Antonovsky). In the present study, it is considered a trait. |
|
2. |
The second important problem of the study is the sample, the participants in the research. The participants are exclusively women of a certain age group. Is there a good reason for such a choice? The author should state the reason for choosing such a specific group. |
The presented work is a part of a larger research project conducted on a group of perimenopausal women (45 - 55 years old). Further studies, including other age groups, are planned. |
|
3. |
The third issue is related to the statistical analysis of the manuscript to which mediation analysis must be added in order to support a model of possible causation. Please also consider whether the sample meets the requirements for regression analysis. |
The author of this paper consulted experienced statisticians (employees of the Faculty of Mathematics and Computer Science of the University of Warmia and Mazury in Olsztyn) who have been dealing with biostatistics for many years. The selection of statistical tests and the correctness of the analyses were verified. In accordance with the reviewer's instructions, extensive changes were made to the manuscript consisting of determining the mediating role of the general sense of coherence and its components in the relationship between global self-esteem and the intensity of perceived stress among the studied nurses. A mediation analysis was performed. Mediation effects were measured according to the methodology of R.M. Baron and D.A. Kenny using regression equations with three steps. A Sobel test was used to test the significance of mediation. |
|
4. |
In closing, I would like to ask for some minor corrections: 1. Delete some information that seems irrelevant to the study such as: Line 148-150 (About 3/4 of the respondents had a gynaecological check-up once a year, whereas the others reported that they had not visited a gynaecologist for a check-up for over two years). Line 162 (frequency of gynaecological check-ups and forms of physical exercise). 2. Add the time period in which the study was conducted, it matters, especially for stress whether it was conducted before or during the pandemic. |
The indicated rows have been deleted. The subsection "Participants" was moved to the section, "Results" and the sociodemographic characteristics of the respondents are presented in a table.
The period during which the study was conducted was added to the manuscript. |
Reviewer 3 Report
Thank you for the opportunity to review this study entitled “Stress intensity and global self-esteem as variables determining the sense of coherence among Polish nurses aged 45-55 years” (ijerph-1525397). The study involved 176 nurses (Mage = 49.1; SD = 3.1), to explore the associations of perceived stress intensity and the global self-esteem level with the sense of coherence in the sphere of comprehensibility, manageability and meaningfulness.
In my opinion, the research topic is relevant, and the study is interesting. However, there are some minor issues that need to be addressed before the paper will be suitable for publication.
- According to the IJERPH guidelines, the structured abstract should not have headings.
- A small suggestion to improve the abstract is to immediately clarify some characteristics of the sample (Mean age and standard deviation, percentage of males and females) to provide a clear snapshot of the subjects involved in the study.
- Starting the introduction with a subsection right away is a bit confusing for the reader. It may be useful to add a few lines in the initial phase of the introduction to explain the general background of reference.
- Please, rephrase the objectives of the study so that they are clearer (in the different sections of the article, e.g., abstract, introduction etc ..)
- Does the sample only include women? If so, this should be clarified more explicitly and highlighted within limits.
- Lines 141. The symbol ± may be deleted since it can be considered implied.
- The heading for "Research instruments" should be 2.3. The headings of the subsequent scales should be 2.3.1 (2.3.2, 2.3.3 etc..). The heading for "Statistical analysis" should be 2.4.
- Please provide indications of internal consistency in the sample of this research for each scale you used.
- Lines 277. The authors stated "Table 2", but to me, it seems to refer to Table 3
- The conclusions were formulated as if they were "highlights". It would be good to formulate this section in a more discursive way and, in addition to summarizing the main results, highlight the practical implications of the results of the research. Lines 424-428 should be moved here.
- Paper that could be integrated over the article to deepen the discussion, as having similar variables and related to the working context, could be:
- https://doi.org/10.1371/journal.pone.0242402
- https://doi.org/10.1177%2F1069072715599357
- https://doi.org/10.3390/ijerph13050459
- https://doi.org/10.3390/su13169065
You may find them here by inserting the doi here (https://www.doi.org/).
Author Response
Dear Reviewers, Thank you very much for a thorough editorial assessment of my manuscript, positive opinions, as well as the reviewers’ remarks. I used them as an important guide to improving the quality of my paper. The corrections were implemented strictly according to their comments. All changes made in the text are marked in yellow and blue. I have enclosed the re-edited manuscript and cover letter as responses to Reviewers, detailing how I followed their suggestions. Thank you very much for your kind consideration of my paper. Yours sincerely, Ewa Kupcewicz, PhD
|
Lp. |
Reviewer 3 |
Responses
|
|
1.
|
According to the IJERPH guidelines, the structured abstract should not have headings. A small suggestion to improve the abstract is to immediately clarify some characteristics of the sample (Mean age and standard deviation, percentage of males and females) to provide a clear snapshot of the subjects involved in the study. Starting the introduction with a subsection right away is a bit confusing for the reader. It may be useful to add a few lines in the initial phase of the introduction to explain the general background of reference. Please, rephrase the objectives of the study so that they are clearer (in the different sections of the article, e.g., abstract, introduction etc ..) |
The abstract of the paper was modified and adapted to IJERPH requirements. The title of the paper was changed and the purpose of the paper was modified. Efforts were made to make the "Introduction" section more valuable. An in-depth literature analysis was carried out and the interpretation of the examined variables (global self-esteem, sense of coherence and stress intensity) was clarified in the "Introduction" section.
|
|
2. |
Does the sample only include women? If so, this should be clarified more explicitly and highlighted within limits. |
The presented work is a part of a larger research project conducted on a group of perimenopausal women (45 - 55 years old). Further studies, including other age groups, are planned. |
|
1. |
Lines 141. The symbol ± may be deleted since it can be considered implied.
|
The “±” symbol was deleted. |
|
2. |
The heading for "Research instruments" should be 2.3. The headings of the subsequent scales should be 2.3.1 (2.3.2, 2.3.3 etc..). The heading for "Statistical analysis" should be 2.4. |
The numbering of sections and subsections has been checked and modified. |
|
3. |
Please provide indications of internal consistency in the sample of this research for each scale you used. |
Internal consistency indicators of the scales used are given. |
|
4. |
Lines 277. The authors stated "Table 2", but to me, it seems to refer to Table 3 |
The subsection, “An analysis of the correlation between the stress intensity, the global self-esteem and the sense of coherence in the study group” was removed. |
|
5. |
The conclusions were formulated as if they were "highlights". It would be good to formulate this section in a more discursive way and, in addition to summarizing the main results, highlight the practical implications of the results of the research. Lines 424-428 should be moved here. |
The subsections, “Limitations and implications for professional practice” and “Conclusions” were modified. |
|
6. |
Paper that could be integrated over the article to deepen the discussion, as having similar variables and related to the working context, could be: https://doi.org/10.1371/journal.pone.0242402 https://doi.org/10.1177%2F1069072715599357 https://doi.org/10.3390/ijerph13050459 https://doi.org/10.3390/su13169065 You may find them here by inserting the doi here (https://www.doi.org/).
|
Some of the proposed literature items were introduced into the manuscript. |
Reviewer 4 Report
It is very interesting topic with the practical application. However, this research is based on one location in Poland - Olsztyn - in the Warmińsko-Mazurskie Voivodship (Poland), so it pretends to be East Poland therefore the application and conclusion can be limited only to this location which could have influence - in fact - on the many psychological aspect of nurses works etc.
Characteristics of analyzed nurses should be presented in the table as it would more clear pictures (lines 141-150).
Even, the Author underlined this limitation - the study group comprised nurses of only one age group- still the Author could present the age structure of nurses in Poland and the percentage of nurses of 45-55 age in all Poland. It would be interesting to get know sth more on this working group in Poland and in this way to get idea on the level of representativeness of this research.
Author should provide also information on the present situation regarding to existence of any programs or not , which are offered for nurses in purpose to create better psychological environmental of their work. Are they or not. It should be explained. The conclusion is too general that some programs can be implemented but it should be formulated also based on the knowledge / information on present situation etc. A kind of inventory of existence or not (also in Olsztyn hospital) of any programs should be provided.
Author Response
Dear Reviewers, Thank you very much for a thorough editorial assessment of my manuscript, positive opinions, as well as the reviewers’ remarks. I used them as an important guide to improving the quality of my paper. The corrections were implemented strictly according to their comments. All changes made in the text are marked in yellow and blue. I have enclosed the re-edited manuscript and cover letter as responses to Reviewers, detailing how I followed their suggestions. Thank you very much for your kind consideration of my paper. Yours sincerely, Ewa Kupcewicz, PhD
|
Lp. |
Reviewer 4 |
Responses
|
|
1. |
It is very interesting topic with the practical application. However, this research is based on one location in Poland - Olsztyn - in the Warmińsko-Mazurskie Voivodship (Poland), so it pretends to be East Poland therefore the application and conclusion can be limited only to this location which could have influence - in fact - on the many psychological aspect of nurses works etc.
Characteristics of analyzed nurses should be presented in the table as it would more clear pictures (lines 141-150).
Even, the Author underlined this limitation - the study group comprised nurses of only one age group- still the Author could present the age structure of nurses in Poland and the percentage of nurses of 45-55 age in all Poland. It would be interesting to get know sth more on this working group in Poland and in this way to get idea on the level of representativeness of this research.
Author should provide also information on the present situation regarding to existence of any programs or not , which are offered for nurses in purpose to create better psychological environmental of their work. Are they or not. It should be explained. The conclusion is too general that some programs can be implemented but it should be formulated also based on the knowledge / information on present situation etc. A kind of inventory of existence or not (also in Olsztyn hospital) of any programs should be provided.
|
The subsection “Participants" was moved to the section “Conclusions" and the sociodemographic characteristics of the respondents are presented in a table.
The presented study is a part of a larger research project conducted on a group of perimenopausal women (45 - 55 years old). Further studies, including other age groups, have been planned.
At the planning stage of the study, a decision was made on the size of the sample population, taking into account statistical requirements as well as technical and organisational aspects. The report on the current staffing situation of nurses and midwives presented by the Head Council of Nurses and Midwives shows that in the Warmińsko-Mazurskie Voivodship in 2021, there were 1864 nurses employed in the 41-50 age range (as of 24 May 2021) (https://www.oipip.olsztyn.pl/wp-content/uploads/2021/12/2021-7-12.pdf). A review of the available literature clearly indicates that there are only a small number of studies taking into account global self-esteem and sense of coherence in stress prevention in the work environment of Polish nurses. Currently, stress prevention programs are being implemented in hospitals located in Olsztyn due to the Covid-19 pandemic, and it is a good time to offer nurses workshop activities aimed at strengthening personal resources, including self-esteem and a sense of coherence. |
Round 2
Reviewer 2 Report
Serious corrections have been made to the manuscript making it of excellent quality in terms of statistical analysis. I would suggest publishing it.
Reviewer 4 Report
There is improvement. Author answered all comments.